

# Fatty acid biomarkers reveal the interaction between two highly migratory species in the Southern Humboldt System: the swordfish and its prey, the jumbo squid

Marco Quispe-Machaca[1], Fabián Guzmán-Rivas[1],
Patricio Barría Martínez[2], Christian Ibáñez[3] and Ángel Urzúa[4,5]

[1] Programa de Doctorado en Ciencias con mención en Biodiversidad y Biorecursos, Facultad de Ciencias., Universidad Católica de la Santísima Concepción, Concepción, Biobío, Chile
[2] IFOP, Instituto de Fomento Pesquero, Valparaíso, Chile
[3] Departamento de Ecología y Biodiversidad, Facultad de Ciencias de la Vida, Universidad Nacional Andrés Bello., Santiago, Metropolitana, Chile
[4] Departamento de Ecología, Facultad de Ciencias, Universidad Católica de la Santísima Concepción, Concepción, Biobío, Chile
[5] Centro de Investigación en Biodiversidad y Ambientes Sustentables (CIBAS), Universidad Católica de la Santísima Concepción, Concepción, Biobío, Chile

Corresponding author
Ángel Urzúa, aurzua@ucsc.cl

## ABSTRACT

Marine trophodynamics refer to the transfer of energy from prey to predators. In marine invertebrates and fishes, the nutrients obtained through the consumption of food and/or prey are stored as energy reserves in certain tissues and/or organs including the liver, muscle, or gonads, and that these are subsequently used as bioenergetic fuel for highly energy-demanding fundamental physiological processes. In the southern Humboldt Current System, the interaction between two highly migratory resources and top species has been observed: the swordfish (*Xiphias gladius*) and its prey the jumbo squid (*Dosidicus gigas*). Because of this trophic interaction, these species store large amounts of energy (as lipids and fatty acids) in their main organs. However, how the fatty acid profile varies in the various organs of the predator and its prey is still unknown, as is its potential use as trophic biomarkers and the ecophysiological role it plays. Our results showed a moderate similarity between the fatty acid profile of the digestive gland of *D. gigas* with the profiles of the liver, gonad, and muscle of *X. gladius*, particularly with fatty acids: palmitic ($C16:0$), stearic ($C18:0$), oleic ($C18:1n9$), gadoleic ($C20:1$), EPA ($C20:5n3$), and DHA ($C22:6n3$). Our findings on the use of fatty acids as biomarkers of the interaction between two highly migratory species in the southern Humboldt System may reveal the degree of preference swordfish have for preying on jumbo squid, particularly through the consumption of the digestive gland. In both species, a high bioenergetic fuel content characterized by a predominance of saturated, monounsaturated, and polyunsaturated fatty acids may be necessary to sustain the high energy costs involved in their migratory and reproductive processes in the Humboldt Current system.

## INTRODUCTION

Trophic marine ecology describes predator *vs*. prey interactions of heterotrophic organisms occurring at different levels of the marine food webs, and at the same trophic level (primary, secondary, top), and especially in the nutrients transfer between and within levels (*Yodzis, 2001*; *Kainz et al., 2017*). In this context, several studies related to understanding trophic ecology are based on: (i) direct observation in the field (*Gaglio et al., 2017*; *Lauriano et al., 2007*), (ii) analyses of stomach contents (*Ng et al., 2021*), (iii) collection of feces from predators (*Trites & Joy, 2005*), (iv) determination of body remains of prey such as bones, otoliths, exoskeletons, and (v) analyses of fatty acid (FAs) profiles and isotopes as biomarkers (*Segura-Cobeña et al., 2021*; *Lazo-Andrade et al., 2021*; *Quispe-Machaca et al., 2022*). FAs have recently been used as biomarkers of the trophic interactions and feeding habits of marine organisms from temperate and cold waters (*Budge, Iverson & Koopman, 2006*; *Galloway & Budge, 2020*; *Góra, Szlinder-Richert & Kornijów, 2022*; *Jardine, Galloway & Kainz, 2020*; *Komisarenko et al., 2021*; *Li et al., 2023*). This has enabled us to explore the trophodynamics of these species in their environments, extending the temporal and spatial scales of these research topics (*Budge, Iverson & Koopman, 2006*; *Graeve & Greenacre, 2020*; *Lazo-Andrade, Barría & Urzúa, 2024*).

In turn, from the stoichiometric and ecophysiological perspective, FAs are important at the structural (as carbon and hydrogen molecules) and functional level (as energy reserves) of organisms (*Galloway & Budge, 2020*). In particular, their greatest importance in marine animals is linked to long-chain (LC) polyunsaturated essential fatty acids (PUFAs) (LC-PUFAs: EPA (C20:5n3), DHA (C22:6n3)), which cannot be biosynthesized *de novo* by organisms, and therefore, must be obtained from the diet and/or consumption of prey that have these LC-PUFAs available in their bodies. Subsequently, these essential biomolecules are conservatively stored in the predators' tissues and are involved in fundamental physiological processes (growth and reproduction) that require energy (*Jardine, Galloway & Kainz, 2020*; *Góra, Szlinder-Richert & Kornijów, 2022*). In this bioenergetic context, it has recently been reported that the relationship of FAs between the predator and its potential prey increases with taxonomic specificity (*Jardine, Galloway & Kainz, 2020*), that is, predators may present a degree of trophic specialization characterized by the consumption of prey with high fat contents, as occurs in the trophic interaction between the jumbo squid (*D. gigas*) and its prey, the red squat lobster (*G. monodon*) (*Quispe-Machaca et al., 2022*). Together, these analyses addressing multidisciplinary aspects (biochemical, physiological, and ecological) allow us to more clearly understand the interaction between the predator and the prey, considering a spatial and temporal scale with a holistic view of the total environment (*Libralato et al., 2014*; *Graeve & Greenacre, 2020*).

In the marine food webs, active predation can be categorized according to the degree of choice and search for prey as follows: (i) selective, large species that eat specific prey with high energy contents (energy maximizer), but in low quantities and (ii) non-selective generalists that consume large amounts of various prey with lower energy contents (time maximizer) (*Jaksic & Marone, 2007*). In an energetic context, the optimal foraging theory

has been used to explain or predict the diet of species related to election of prey with high energetic contend (*Spitz et al., 2010*). In particular, active predation strategies describe how marine organisms maximize the energy they obtain from prey and minimize the time spent during the entire feeding process, from the consumption of prey, its ingestion and finally the assimilation of nutrients. In this context it is also important to consider that organisms can also carry out other energy-demanding physiological and behavioral activities while feeding, such as tolerating the exposure to variations of abiotic factors such as temperature, oxygen, and salinity (*Carrier-Belleau et al., 2021*) and behaviors such as escaping from predators, searching for a mate, among others (*Kie, 1999*). The optimal foraging theory also considers that the nutrients obtained through the consumption of food and/or prey are stored as energy reserves (*i.e.*, FAs) in certain tissues including the liver, muscle, or gonads, and that these are subsequently used as bioenergetic fuel for highly energy-demanding fundamental physiological processes, such as growth and reproduction (*Dannenberger et al., 2020*; *Jay, Iverson & Fischbach, 2021*).

Species of higher trophic level, mostly described as strict carnivores, sustain their lifestyle and high energy demands by consuming only certain prey with high carbon and nitrogen contents (*i.e.*, lipids and proteins, respectively), and also feed only specific parts of the prey (*i.e.*, energy storage organs: livers, hepatopancreas, and digestive glands) that they capture through the use of specialized mouth structures (*Berkovitz & Shellis, 2017*; *Preti et al., 2023*). In the context of this energy maximization strategy, the same trend has been described in terrestrial (*e.g.*, lions capture and consume only the liver of their prey) (*Sibly et al., 2013*), and also in marine environments (*Massing et al., 2022*; *Lazo-Andrade, Barría & Urzúa, 2024*). Particularly, here the swordfish slashes the jumbo squid with its sword and mainly consumes its digestive gland, probably because of its high fat content, which has been recorded in the field by scientific observers (*Markaida & Hochberg, 2005*; *Barría et al., 2024*; *SUBPESCA-Subsecretaria de Pesca y Acuicultura, 2024*). Considering the fact that only the presence of cuts on the body of their prey has been published, presumably due to the use of the sword (*Abid & Idrissi, 2006*; *FishBase, 2024*), it is necessary to carry out detailed comparative studies that use FAs as trophic biomarkers, consider the same space-time window (*i.e.*, a simultaneous comparison) and include the interaction of these two fishery resources considered as "higher level species" and "highly migratory" that interact with one another in the southern Humboldt System, such as the swordfish and its prey, the jumbo squid.

In the Humboldt Current System (HCS), considered one of the most important coastal upwelling ecosystems on the planet (*Montecino & Lange, 2009*), the model species studied (jumbo squid: *D. gigas* and swordfish *X. gladius*) are considered highly migratory resources (*Hu et al., 2022*; *Quispe-Machaca et al., 2021*; *Stewart et al., 2014*; *Su et al., 2020*) that interact trophically with one another (*i.e.*, *D. gigas* as the main prey of swordfish; *Castillo et al., 2007*; *Ibáñez, González & Cubillos, 2004*; *Letelier et al., 2009*). Due to the highly migratory activity of swordfish, this species requires large amounts of energy provided by food, which is stored in its organs (liver, muscle, gonad) in the form of lipids and FAs to support its fundamental physiological processes of maintenance, growth, and reproduction (*Lazo-Andrade et al., 2021*). Numerous studies analyzing the stomach
content of swordfish in HCS have indicated that their diet is mainly based on cephalopods (*Castillo et al., 2007*; *Ibáñez, González & Cubillos, 2004*; *Letelier et al., 2009*; *Yáñez et al., 1996*). Despite the capture restrictions of these species (event sampling in a narrow time window), these studies indicate how cephalopods (in this case *D. gigas*) as food and/or prey items can be an important source of energy for the swordfish *X. gladius* (*Quispe-Machaca et al., 2021*; *Lazo-Andrade et al., 2021*). However, details regarding how lipids and FAs vary in the organs of the prey and predator are still unknown, as is their potential use as trophic biomarkers, which could reveal more about their dynamics, and the potential physiological role that lipids and fatty acids play. In this context, identifying the FAs profiles of different organs of the jumbo squid could therefore provide key information on the rate of assimilation of these essential nutrients and how these are subsequently transferred and stored conservatively in the different organs of its predator *X. gladius* (*Gong et al., 2020*; *Quispe-Machaca et al., 2021*) to then be used as bioenergetic fuel in various key physiological processes of the energy balance model (homeostasis, reproduction, growth).

As previously mentioned, to date studies have focused on intra-individual variations of FAs, but not on the simultaneous interaction among them (*i.e.*, comparison of FAs between tissues and species). Therefore, this is the first study that shows comparative results of the FAs of the predator and its prey, captured on the same space-time scale. Due to the selective feeding habit of the swordfish on the organs (*e.g.*, digestive gland) of the jumbo squid, and its low capacity to biosynthesize FAs, these biomolecules are expected to be conservatively incorporated and stored in swordfish tissues. Consequently, some degree of similarity in the FA profiles found in the tissues of the predator and its prey is expected. Consequently, the objective of this study was to evaluate, using FAs analyses as suitable tracers, trophic interactions and ecophysiological aspects of two highly migratory resources that coincide in one of the most productive marine ecosystems on the planet, the Humboldt Current System.

## MATERIALS AND METHODS

### Ethical declaration

This research was conducted in accordance with the Act on Welfare and Management of Marine Animals, and they comply with the current Chilean animal care and manipulation legislation of the fishery resources (*SUBPESCA-Subsecretaria de Pesca y Acuicultura, 2024*). Consequently, to avoid the pain of the animals during their capture and processing, they were euthanized with a thermal shock of rapid freezing ($-20\ °C$) (Law 20.380; Ministry of Health and Ethics Committee, Chile) (*Robb & Kestin, 2002*).

### Capture, sample processing and transport to the laboratory

As part of the monitoring program of "Fisheries Project of Highly Migratory Resources: Biological-Fishing Aspects, 2022" of Instituto de Fomento Pesquero (IFOP), adult specimens of *D. gigas* ($N = 32$) and *X. gladius* ($N = 31$) were captured in the same fishing area where these species coincide temporally and spatially in the southern Humboldt System (Autumn-Winter: 34°–36°S & 73°–76°W) (*Lazo-Andrade et al., 2021*; *Quispe-Machaca et al., 2022*). It is important to mention that, due to the low availability of

specimens in the field, restrictive capture for this study area (catch by jigs) and fishing ban periods (*SUBPESCA-Subsecretaria de Pesca y Acuicultura, 2024*), it was only possible to capture specimen samples in their habitat (far from the coast) for a narrow window of time. This permitted only opportunity sampling once a year.

On board, the animals were measured (jumbo squid: 66 ± 5.07 cm of mantle length; swordfish: 216 ± 16.31 cm of lower jaw fork length), sexed and dissected following the methodology described by *Quispe-Machaca et al. (2021)* for jumbo squid, and for swordfish by *Lazo-Andrade et al. (2021)*. Five to 10 g of fresh tissue samples were extracted from the organs (jumbo squid: digestive gland (32), gonad (32), mantle muscle (32); swordfish: liver (three), gonad (three), muscle (31)) and preserved in 250 mL thermo-hermetic plastic flasks using dry ice. Subsequently, the samples were transported in hermetic boxes with dry ice to the Laboratory of Hydrobiological Resources of the Universidad Catolica de la Santisima Concepcion de Chile, where they were cold homogenized, sonicated and then dried and/or exposed to lyophilization (lyophilizer FDU-7012 Operon, for 48 h at −80 °C). Afterwards, a 20–30 mg dry weight (DW) tissue sample of each organ was extracted for fatty acid profile analyses.

## Fatty acid profiles

First, the total lipid content was obtained following the methodology described by *Cequier-Sánchez et al. (2008)* and recently applied for highly migratory marine species (for details of procedures, please see: *Quispe-Machaca et al., 2022*; *Lazo-Andrade et al., 2023*). For this, a 20 mg dry weight tissue sample was weighed with a Sartorius digital analytical balance (d = 0.1 mg), maintained at a constant cold temperature (ca. 5 °C) and immersed in 5 mL of dichloromethane:methanol (2:1) solution for 12 h. Then, it was sonicated (AC-120H equipment, MRC) at the same cold temperature for 15 min, after which 4 mL of potassium chloride (0.88% KCl in ultra-pure water) was added and centrifuged at 1,500 RPM for 5 min (FASCIO TG1650-S). Afterwards, the lower phase (*i.e.*, containing the total lipids) was extracted and transferred to amber vials that had been previously weighed (*i.e.*, dried empty weight). Finally, the total lipids were quantified by evaporating the solvent (*i.e.*, with an injection of nitrogen gas) in a sample concentrator (109A YH-1; Glas-Col) and subtracting the empty dry weight of each vial from the new dry weight of the vials containing the extracted lipids.

The fatty acid profiles of the samples were determined following the method used by *Malzahn et al. (2007)*. Thus, the fatty acid methyl esters (FAMEs) of each sample were assessed using the previously quantified total lipid extracts. In brief, 1 mL of the lipid extract was esterified by incubating them in methanolic sulfuric acid at 70 °C for 1 h in a Thermo-Shaker (MRC model DBS-001). After that, the fatty acids were rinsed three times with 6 mL, 3 mL, and 3 mL of n-hexane, and concentrated to 1 mL in amber vials using a sample concentrator (MD 200) and nitrogen. FAMEs were measured with a gas chromatograph (GC, Agilent, model 7890 A) equipped with a DB-225 column (J and W Scientific, 30 m long, 0.25 intermediate diameter, and 0.25 mm film). We used the temperature program for sample injections suggested for the GC column. Briefly, the oven temperature was set at 100 °C for 4 min and increased 3 mL/min to 240 °C for 15 min. The

individual FAMEs were identified by comparing them to known standards of fatty acids of marine origin (certified material, Supelco 37 FAME mix 47885-U; *Malzahn et al., 2007*). Using chromatography software (ChemStation; Agilent), FAMEs were quantified by means of the response factor to the internal standard blank (corresponding to a C23:0 fatty acid added prior to transmethylation) (*Malzahn et al., 2007*; *Urzúa & Anger, 2011*).

## Statistical analysis

To evaluate differences in total fatty acids (saturated, SFA; monounsaturated, MUFA; polyunsaturated, PUFA) among organs for each species, a Kruskall-Wallis (H) test was performed. In turn, to compare the fatty acid profiles of *X. gladius* and its prey *D. gigas*, the fatty acids that contributed at least 5% of the total weight were selected, and their concentration values were transformed into % and $\log (X + 1)$ (for details see: *Pethybridge et al. (2013)*). In turn, a Bray-Curtis resemblance matrix was performed to develop a multivariate principal coordinate analysis (PCoA). The fatty acid profiles detected in the different organs of the prey *D. gigas* and its predator *X. gladius* were compared with a PCoA. For the observed differences, a PERMANOVA was performed. In this test it is not necessary to comply with the required assumptions compared to parametric tests such as ANOVA (*Anderson, 2017*) with a significance level of 0.05. In addition, a similarity analysis (ANOSIM) was performed to determine similar groups with a coefficient of determination close to zero (R = 0) and highly dissimilar groups close to one (R = 1) (*Somerfield, Clarke & Gorley, 2021*) to statistically verify the dissimilarities that occur between the tissues observed through PERMANOVA. Subsequently, a similarity percentage analysis (SIMPER) was carried out to identify the percentage of contribution of fatty acids, which contributed to the differences observed in the studied organs (*Graeve & Greenacre, 2020*). All multivariate analyses were performed with Primer V6 software (*Clarke & Gorley, 2015*).

## RESULTS

### Fatty acid profile of *D. gigas*

The highest concentration of total fatty acids (SFA, saturated fatty acid; MUFA, monounsaturated fatty acid; PUFA, polyunsaturated fatty acid) was found in the digestive gland, followed by the gonad and muscle (Table 1). Significant differences were observed between tissues in each of the total fatty acid classes (K-W: SFA: $H(2,96) = 76.098$, $p < 0.05$, MUFA: $H(2,96) = 81.4899$, $p < 0.05$; ANOVA: PUFA: $F(2,93) = 177.131$, $p < 0.05$) (Fig. 1A). N6 polyunsaturated fatty acids (n6 PUFA) were only present in the digestive gland, which presented the greatest diversity of fatty acids ($N = 26$ total; saturated fatty acids, SFA: $N = 10$; monounsaturated fatty acids, MUFA: $N = 7$; polyunsaturated fatty acids, PUFAn6: $N = 5$, PUFAn3: $N = 4$).

The digestive gland presented a total content of PUFA of $144.95 \pm 30.97$ mg FA g DW$^{-1}$, with a higher percentage, 27.54% of docosahexaenoic (DHA, C22:6n3) and 14.69% of eicosapentaenoic (EPA, C20:5n3) (Table 1). The total content of MUFA found was $71.15 \pm 12.06$ mg FA g DW$^{-1}$, the fatty acids with the highest percentage were oleic (C18:1n9) with 10.84%, gadoleic (C20:1) with 4.84% and palmitoleic (C16:1) with 4.34% (Table 1). The

**Table 1 Profile of fatty acids of *Dosidicus gigas* present in the digestive gland, gonad and muscle expressed in mg FA g DW$^{-1}$.**

|  | Fatty acid (FA) | Digestive gland | | Gonad | | Mantle muscle | |
| --- | --- | --- | --- | --- | --- | --- | --- |
|  |  | Mean ± SD | % | Mean ± SD | % | Mean ± SD | % |
| SFA | C8:0 | 0.16 ± 0.04 | 0.05 | 0.22 ± 0.04 | 0.87 | 0.26 ± 0.03 | 1.37 |
|  | C12:0 | 0.37 ± 0.10 | 0.12 | 0.78 ± 0.00 | 3.11 | – | – |
|  | C13:0 | 0.27 ± 0.05 | 0.09 | – | – | – | – |
|  | C14:0 | 14.90 ± 4.23 | 4.86 | 0.28 ± 0.12 | 1.12 | 0.21 ± 0.01 | 1.08 |
|  | C15:0 | 1.68 ± 0.45 | 0.54 | – | – | – | – |
|  | C16:0 | 53.84 ± 13.43 | 17.57 | 4.34 ± 0.96 | 17.38 | 3.53 ± 0.54 | 18.46 |
|  | C17:0 | 2.18 ± 0.50 | 0.71 | 0.41 ± 0.14 | 1.64 | – | 0 |
|  | C18:0 | 12.45 ± 2.80 | 4.06 | 2.36 ± 0.88 | 9.44 | 1.23 ± 0.27 | 6.43 |
|  | C20:0 | 1.43 ± 0.45 | 0.47 | – | – | – | – |
|  | C23:0 | 3.07 ± 1.16 | 1.00 | – | – | – | – |
| TOTAL SFA |  | 90.36 ± 17.12 | 29.49 | 8.38 ± 1.83 | 33.56 | 5.23 ± 1.41 | 27.34 |
| MUFA | C14:1 | 0.31 ± 0.11 | 0.10 | – | – | – | – |
|  | C16:1 | 13.40 ± 4.35 | 4.34 | 0.29 ± 0.00 | 1.17 | 3.91 ± 0.00 | 20.46 |
|  | C17:1 | 2.79 ± 1.64 | 0.91 | – | – | – | – |
|  | C18:1n9 | 33.48 ± 11.15 | 10.84 | 0.65 ± 0.24 | 2.61 | 0.47 ± 0.20 | 2.43 |
|  | C20:1 | 14.95 ± 4.98 | 4.88 | 3.09 ± 0.72 | 12.36 | 1.09 ± 0.17 | 5.68 |
|  | C22:1n9 | 3.80 ± 2.06 | 1.24 | – | – | – | – |
|  | C24:1 | 2.43 ± 0.69 | 0.79 | – | – | – | – |
| TOTAL MUFA |  | 71.15 ± 12.06 | 23.22 | 4.03 ± 1.35 | 16.14 | 5.46 ± 0.56 | 28.57 |
| PUFA n6 | C18:2n6t | 2.62 ± 0.84 | 0.85 | – | – | – | – |
|  | C18:3n6 | 1.00 ± 0.37 | 0.33 | – | – | – | – |
|  | C20:2n6 | 2.97 ± 0.89 | 0.97 | – | – | – | – |
|  | C20:3n6 | 0.94 ± 0.20 | 0.31 | – | – | – | – |
|  | C20:4n6/C21:0 | 1.02 ± 0.19 | 0.33 | – | – | – | – |
| TOTAL PUFA n6 |  | 8.55 ± 1.13 | 2.79 | – | – | – | – |
| PUFA n3 | C18:3n3 | 3.47 ± 1.07 | 1.13 | – | – | – | – |
|  | C20:3n3 | 3.49 ± 1.04 | 1.14 | 0.91 ± 0.28 | 3.65 | – | – |
|  | C20:5n3 | 45.03 ± 17.36 | 14.69 | 5.48 ± 2.46 | 21.95 | 2.32 ± 0.77 | 12.13 |
|  | C22:6n3 | 84.41 ± 27.84 | 27.54 | 6.17 ± 2.84 | 24.7 | 6.11 ± 2.32 | 31.96 |
| TOTAL PUFA n3 |  | 136.40 ± 37.12 | 44.51 | 12.57 ± 3.16 | 50.3 | 8.43 ± 2.59 | 44.09 |
| TOTAL PUFA |  | 144.95 ± 30.97 | 47.30 | 12.57 ± 3.16 | 50.3 | 8.43 ± 2.59 | 44.09 |
| TOTAL FA |  | 306.46 ± 22.09 | 100 | 24.98 ± 2.57 | 100 | 19.12 ± 2.22 | 100 |

Note:
Mean values ± standard deviation (SD), %, percentage of fatty acid; SFA, saturated fatty acid; MUFA, monounsaturated fatty acid; PUFA, polyunsaturated fatty acid.

total content of SFA was 90.36 ± 17.12 mg FA g DW$^{-1}$, the highest percentages of fatty acids were palmitic (C16:0) with 17.57%, myristic (C14:0) with 4.86% and stearic (C18:0) with 4.06% (Table 1).

In turn, the gonad presented a PUFA content of 12.57 ± 3.16 mg FA g DW$^{-1}$. The fatty acids with the highest percentage in the gonad were DHA (C22:6n3) with 24.7% and EPA

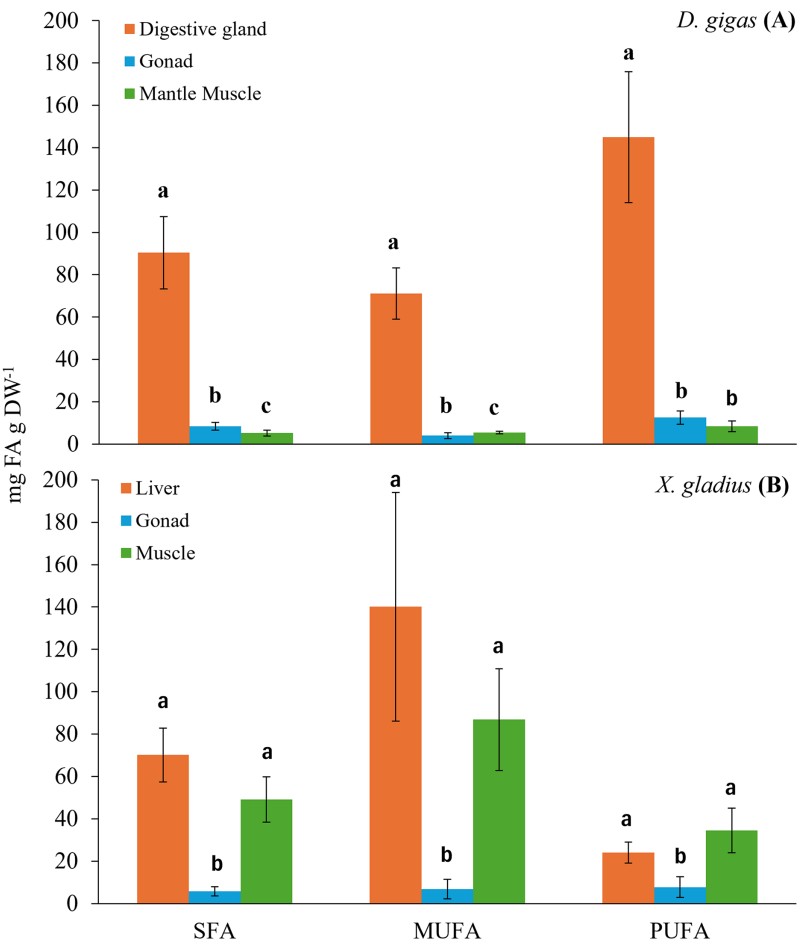

**Figure 1** Bar chart of the total fatty acid classes (SFA: saturated; MUFA: monounsaturated; PUFA: polyunsaturated) present in the organs of the prey (A) *D. gigas*, and its predator (B) (*X. gladius*). Letters represent statistically significant differences at 95%.

(C20:5n3) with 21.95% (Table 1). The total MUFA content was $4.03 \pm 1.35$ mg FA g $DW^{-1}$, and gadoleic (C20:1) showed the highest percentage with 12.36% (Table 1). The total SFA content was $8.38 \pm 1.83$ mg FA g $DW^{-1}$, and the SFA with the highest percentage was palmitic (C16:0) with 17.38% (Table 1).

In the muscle, the total PUFA content was $8.43 \pm 2.59$ mg FA g $DW^{-1}$, only two n3 fatty acid chains were observed, including DHA (C22:6n3) with 31.96% and EPA (C20:5n3) with 12.13% (Table 1). The total MUFA content was $5.46 \pm 0.56$ mg FA g $DW^{-1}$, the MUFAs with the highest percentage were palmitoleic (C16:1) with 20.46% and gadoleic (C20:1) with 5.68% (Table 1). The total SFA content was $5.23 \pm 1.41$ mg FA g $DW^{-1}$, and the SFA with the highest percentage was palmitic (C16:0) with 18.46% (Table 1).

## Fatty acid profile of *X. gladius*

The evaluation of the fatty acids found in the different organs of the swordfish *X. gladius* resulted in the liver being the organ with the highest total content of fatty acids, followed by muscle, and the gonad showed the lowest fatty acid content (Table 2). Statistically

**Table 2 Fatty acid profile of the liver, gonad, and muscle of *Xiphias gladius* expressed in mg FA g DW$^{-1}$.**

|  | Fatty acid (FA) | Liver | | Gonad | | Muscle | |
|---|---|---|---|---|---|---|---|
|  |  | Mean ± SD | % | Mean ± SD | % | Mean ± SD | % |
| SFA | C12:0 | – | – | 0.44 ± 0.00 | 2.18 | 0.10 ± 0.05 | 0.06 |
|  | C13:0 | – | – | – | – | 0.15 ± 0.09 | 0.09 |
|  | C14:0 | 2.32 ± 1.31 | 0.99 | 0.33 ± 0.34 | 1.61 | 4.54 ± 2.26 | 2.66 |
|  | C15:0 | 0.43 ± 0.13 | 0.18 | 0.21 ± 0.06 | 1.04 | 1.00 ± 0.73 | 0.58 |
|  | C16:0 | 23.98 ± 9.48 | 10.24 | 2.22 ± 3.21 | 10.96 | 26.07 ± 13.95 | 15.29 |
|  | C17:0 | 1.62 ± 0.01 | 0.69 | 0.28 ± 0.16 | 1.39 | 1.43 ± 1.08 | 0.84 |
|  | C18:0 | 22.19 ± 15.62 | 9.48 | 1.09 ± 1.64 | 5.42 | 12.40 ± 7.04 | 7.27 |
|  | C20:0 | 0.67 ± 0.13 | 0.29 | – | – | 0.56 ± 0.48 | 0.33 |
|  | C21:0 | – | – | 0.18 ± 0.00 | 0.9 | 0.68 ± 0.00 | 0.4 |
|  | C22:0 | – | – | – | – | 0.40 ± 0.00 | 0.23 |
|  | C23:0 | 3.23 ± 2.33 | 1.38 | 0.79 ± 0.49 | 3.93 | 1.56 ± 1.29 | 0.91 |
|  | C24:0 | 15.64 ± 21.73 | 6.68 | 0.20 ± 0.08 | 1.00 | 0.23 ± 0.12 | 0.14 |
| TOTAL SFA | | 70.08 ± 12.78 | 29.93 | 5.75 ± 2.10 | 28.44 | 49.12 ± 10.65 | 28.8 |
| MUFA | C14:1 | 0.08 ± 0.00 | 0.03 | 0.09 ± 0.03 | 0.45 | 0.15 ± 0.10 | 0.09 |
|  | C15:1 | – | – | – | – | 0.41 ± 0.46 | 0.24 |
|  | C16:1 | 5.72 ± 2.93 | 2.44 | 0.49 ± 0.59 | 2.4 | 6.47 ± 2.30 | 3.8 |
|  | C17:1 | 1.78 ± 0.62 | 0.76 | 0.27 ± 0.13 | 1.34 | 1.35 ± 0.94 | 0.79 |
|  | C18:1n-9 | 112.91 ± 66.37 | 48.22 | 4.29 ± 7.08 | 21.24 | 65.99 ± 29.32 | 38.69 |
|  | C20:1n-9 | 17.18 ± 11.74 | 7.34 | 0.98 ± 1.24 | 4.84 | 9.21 ± 5.44 | 5.4 |
|  | C22:1n-9 | 2.43 ± 0.22 | 1.04 | 0.38 ± 0.13 | 1.88 | 1.65 ± 1.11 | 0.97 |
|  | C24:1 | – | – | 0.24 ± 0.09 | 1.21 | 1.63 ± 1.03 | 0.95 |
| TOTAL MUFA | | 140.09 ± 54.04 | 59.83 | 6.74 ± 4.60 | 33.36 | 86.85 ± 24.01 | 50.92 |
| PUFA n6 | C18:2n-6t | 1.34 ± 0.82 | 0.57 | 0.22 ± 0.20 | 1.11 | 1.25 ± 0.85 | 0.74 |
|  | C18:2n-6c | 0.60 ± 0.30 | 0.26 | 0.19 ± 0.04 | 0.92 | 0.44 ± 0.35 | 0.26 |
|  | C18:3n-6 | 0.46 ± 0.14 | 0.2 | 0.08 ± 0.00 | 0.42 | 0.27 ± 0.20 | 0.16 |
|  | C20:2n-6 | 0.88 ± 0.11 | 0.38 | 0.20 ± 0.03 | 1.01 | 0.49 ± 0.41 | 0.29 |
|  | C20:4n-6 | – | – | 0.18 ± 0.00 | 0.9 | 0.68 ± 0.00 | 0.4 |
| TOTAL PUFA n6 | | 3.29 ± 0.58 | 1.41 | 0.88 ± 0.16 | 4.36 | 3.14 ± 0.63 | 1.84 |
| PUFA n3 | C18:3n-3 | 0.50 ± 0.20 | 0.21 | 0.28 ± 0.17 | 1.38 | 1.01 ± 0.63 | 0.59 |
|  | C20:3n-3 | 2.09 ± 0.08 | 0.89 | 0.54 ± 0.61 | 2.69 | 1.45 ± 1.02 | 0.85 |
|  | EPA C20:5n-3 | 1.34 ± 0.20 | 0.57 | 1.03 ± 0.16 | 5.1 | 4.00 ± 1.63 | 2.35 |
|  | DHA C22:6n-3 | 16.74 ± 0.75 | 7.15 | 4.99 ± 7.37 | 24.68 | 24.99 ± 13.90 | 14.65 |
| TOTAL PUFA n3 | | 20.68 ± 6.50 | 8.83 | 6.84 ± 5.28 | 33.84 | 31.44 ± 13.75 | 18.44 |
| TOTAL PUFA | | 23.97 ± 4.97 | 10.24 | 7.72 ± 4.91 | 38.2 | 34.51 ± 10.53 | 20.28 |
| TOTAL FA | | 234.15 ± 30.35 | 100 | 20.21 ± 3.91 | 100 | 170.55 ± 16.28 | 100 |

**Note:**
Mean values ± standard deviation (SD), %, percentage of fatty acid; SFA, saturated fatty acid; MUFA, monounsaturated fatty acid; PUFA, polyunsaturated fatty acid.

significant differences were observed among tissues in each class of total fatty acid (ANOVA: SFA: $F(2,34) = 27.765$, $p < 0.05$, MUFA: $F(2,34) = 28.975$, $p < 0.05$, PUFA: $F(2,34) = 9.8180$, $p < 0.05$) (Fig. 1B).

The liver presented the highest total MUFA content with $140.09 \pm 54.04$ mg FA g $DW^{-1}$, the MUFAs with the highest percentages were oleic (C18:1n9) with 48.22% and gondoic (C20:1n9) with 7.34% (Table 2). The SFAs presented a total content of $70.08 \pm 12.78$ mg FA g $DW^{-1}$; the SFAs with the highest percentages were palmitic (C16:0) with 10.24% and Stearic (C18:0) with 9.48% (Table 2). The total PUFA content was $23.97 \pm 4.97$ mg FA g $DW^{-1}$. The n3 PUFA content was higher than the n6 PUFA content, in this organ the arachidonic acid (ARA, C20:4n-6) was not registered in comparison to the other organs (gonad and muscle). Among the n3 PUFAs, the highest percentage was DHA (C22:6n3) with 7.15% (Table 2).

The total fatty acid content in the gonad was $20.21 \pm 3.91$ mg FA g $DW^{-1}$. This organ had a higher content of PUFA with a total of $7.72 \pm 4.91$ mg FA g $DW^{-1}$, of which the n3 PUFA content was higher than the n6 PUFA content, which was low. The n3 PUFAs that presented the highest percentages were the DHA (C22:6n3) fatty acids with 24.68% and EPA (C20:5n3) with 5.10% (Table 2). The total MUFA content in the gonad was $6.74 \pm 4.60$ mg FA g $DW^{-1}$; the most abundant MUFAs were oleic (C18:1n9) with 21.24%, followed by gondoic (C20:1n9) with 4.84% (Table 2). The SFAs presented a total content of $5.75 \pm 2.10$ mg FA g $DW^{-1}$, the most predominant SFAs were palmitic (C16:0) and Stearic (C18:0), each with 10.96% and 5.42%; respectively (Table 2).

In the muscle of *X. gladius*, MUFAs were the most abundant fatty acids with a total content of $170.55 \pm 16.28$ mg FA g $DW^{-1}$, the MUFAs with the highest percentages were oleic (C18:1n9), gondoic (C20:1n9) and palmitoleic (C16:1), with 38.69%, 5.40% and 3.80%, respectively (Table 2). The SFAs presented a total content of $49.12 \pm 10.65$ mg FA g $DW^{-1}$, with a high predominance of palmitic (C16:0) with 15.29% and stearic (C18:0) with 7.27% (Table 2). The PUFAs showed the lowest total content of fatty acids with $34.51 \pm 10.53$ mg FA g $DW^{-1}$. The n3 PUFAs demonstrated a higher total content in relation to the n6 PUFAs. Among the n3 PUFAs, DHA (C22:6n3) was the most abundant fatty acid with 14.65%, followed by EPA (C20:5n3) with 2.35%. In relation to the n6 PUFAs, Linolelaidic (C18:2n6t) presented a low abundance with 0.74% (Table 2).

## A comparison of the fatty acids found in predator *X. gladius* and its prey *D. gigas*

When comparing the FA profiles of the organs of the predator with its prey, differences in the grouping of FA profiles within and between analyzed species were evident (Fig. 2). In particular, the spatial arrangement of FAs was observed in the following groups: (i) the gonad and muscle of the jumbo squid (with the contribution of the FAs: C20:1, C22:6n3, C20:5n3), (ii) the three swordfish tissues with the jumbo squid digestive gland (with C18:1n9), and (iii) a great dispersion in the FA profile of the swordfish gonad was also revealed (with C16:0) (Fig. 2). By means of PERMANOVA, statistically significant differences were observed (PERMANOVA: pseudo-$F_{5,132} = 44.433$, $p = 0.001$) in the different tissues of the predator (*X. gladius*) and its prey (*D. gigas*).

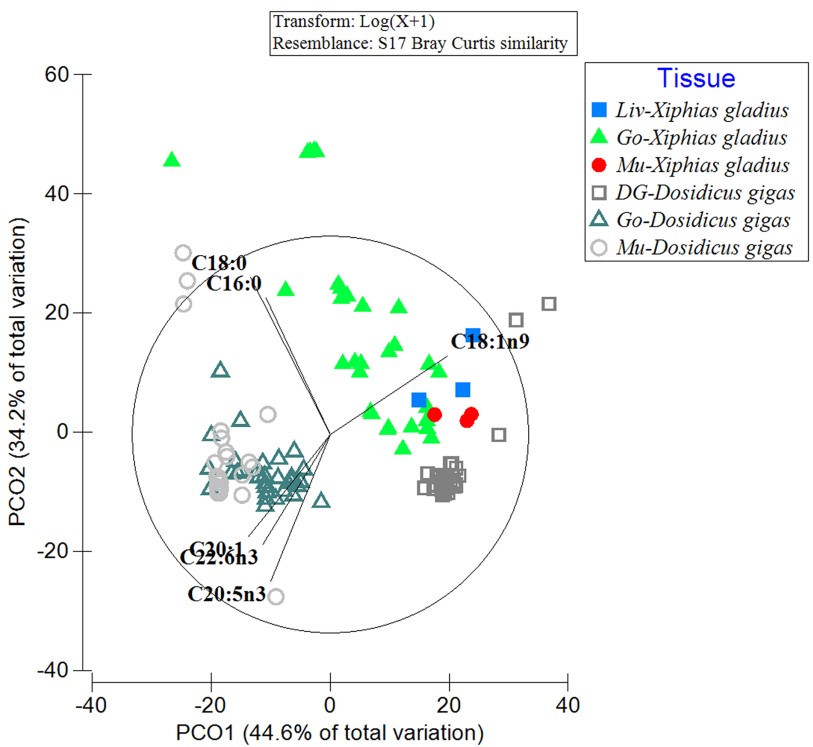

**Figure 2 Comparison of the fatty acid profiles among the organs of the prey *D. gigas* and its predator *X. gladius*, through principal coordinate analysis (PCoA).** Liv, Liver; Go, Gonad; Mu, Muscle and DG, Digestive gland.

In turn, in the ANOSIM, the FAs detected in the liver of *X. gladius* were different from those in the gonad (R-ANOSIM = 0.988) and muscle (R-ANOSIM = 0.979) of its prey *D. gigas*. When comparing the gonad organ of *X. gladius* with the tissues of its prey, the gonad (R-ANOSIM = 0.721) and the mantle muscle (R-ANOSIM = 0.745) of *D. gigas* were found to be very different, while the digestive gland was quite similar to the gonad. Finally, when comparing the muscle profiles of the predator *X. gladius* with the tissue of its prey *D. gigas*, a similarity was observed with the digestive gland (R-ANOSIM = 0.572), while the gonad (R-ANOSIM = 0.996) and mantle muscle (R-ANOSIM = 0.977) were very different (Table 3). In particular, the R values of 0.5 were recorded when comparing FAs of predator tissues with the prey's digestive gland, indicating similarity.

When evaluating the percentage of contribution to the fatty acid profiles by means of SIMPER (Table 4), the fatty acids DHA (C22:6n3), palmitic (C16:0), EPA (C20:5n3), and oleic (C18:1n9) in the digestive gland of *D. gigas* contributed the most according to the similarity evaluation, with a cumulative contribution of 70.39%. In the gonad, the fatty acids that contributed the most were DHA (C22:6n3), EPA (C20:5n3), palmitic (C16:0), gadoleic (C20:1), and stearic (C18:0), with an accumulated percentage of 95.72%. In the muscle, the fatty acids with the greatest contribution were DHA (C22:6n3), palmitic (C16:0), EPA (C20:5n3) and stearic (C18:0); their accumulated percentage was 92.22%. In these three organs, the C22:6n3 polyunsaturated fatty acid (DHA) presented the greatest contribution in the digestive gland (25.63%), gonad (27.51%) and muscle (44.88%).

**Table 3 Analysis of similarity (ANOSIM) and pairwise comparisons among the organs of the prey (*Dosidicus gigas*) and its predator (*Xiphias gladius*).**

| Group | R statistic | Level of significance % |
|---|---|---|
| Gonad-*Xg*, Muscle-*Xg* | 0.119 | 17.8 |
| Gonad-*Xg*, Liver-*Xg* | 0.18 | 13.4 |
| Gonad-*Xg*, Digestive gland-*Dg* | 0.576 | 0.1 |
| Gonad-*Xg*, Gonad-*Dg* | 0.721 | 0.1 |
| Gonad-*Xg*, Muscle-*Dg* | 0.745 | 0.1 |
| Muscle-*Xg*, Liver-*Xg* | 0.296 | 20 |
| Muscle-*Xg*, Digestive gland-*Dg* | 0.572 | 1.3 |
| Muscle-*Xg*, Gonad-*Dg* | 0.996 | 0.1 |
| Muscle-*Xg*, Muscle-*Dg* | 0.977 | 0.1 |
| Liver-*Xg*, Digestive gland-*Dg* | 0.709 | 0.6 |
| Liver-*Xg*, Gonad-*Dg* | 0.988 | 0.1 |
| Liver-*Xg*, Muscle-*Dg* | 0.979 | 0.2 |
| Digestive gland-*Dg*, Gonad-*Dg* | 0.862 | 0.1 |
| Digestive gland-*Dg*, Muscle-*Dg* | 0.874 | 0.1 |
| Digestive gland-*Dg*, Muscle-*Dg* | 0.504 | 0.1 |

**Note:**
Dg, *Dosidicus gigas*; Xg, *Xiphias gladius*.

In all of the evaluated organs (liver, gonad, muscle) of *X. gladius*, the fatty acid that presented the greatest contribution in the percentage of similarity was oleic (C18:1n9). The fatty acids with the greatest contribution in the liver were oleic (C18:1n9), palmitic (C16:0) and stearic (C18:0), with an accumulated percentage of 81.41%; DHA (C22:6n3) polyunsaturated fatty acids (2.83%) presented a low percentage contribution in this organ. In the gonad, the fatty acids oleic (C18:1n9), palmitic (C16:0), DHA (C22:6n3) and stearic (C18:0) had a high contribution, with an accumulated percentage of 93.28%. The fatty acids with the greatest contribution in the muscle were Oleic (C18:1n9), palmitic (C16:0) and DHA (C22:6n3), with an accumulated percentage of 71.64%. In turn, in this last organ the monounsaturated fatty acids (palmitoleic (C16:1), gadoleic (C20:1)) showed a low contribution.

## DISCUSSION

This study explored the trophic interaction (predator *vs.* prey) between two highly migratory resources of the Humboldt System (jumbo squid and swordfish prey) by comparing the fatty acid profiles of their tissues and/or organs. Our findings revealed that analyzing the fatty acids used as trophic biomarkers among these species could be a useful tool that may reflect the predator dietary intake. In our study, and as reported for several species of marine animals classified as large predators (seals: *Lobodon carcinophaga, Leptonychotes weddellii, Hydrurga leptonyx* (Guerrero et al., 2021); white shark *Carcharodon carcharias* (Pethybridge et al., 2014); fishes: *Kajikia audax, Makaira nigricans, Coryphaena hippurus* (Young et al., 2018)), the lipids and consequently the key fatty acids (precursors: oleic acid (C18:1n9), linoleic (C18:2n6), gamma-linolenic

**Table 4 Analysis of the percentage of similarity (SIMPER) of the fatty acids present in the organs of the digestive gland, gonad and muscle of the prey *Dosidicus gigas* and its predator *Xiphias gladius*.**

| Species | Tissue | Average similarity % | Fatty acid | Average Abun | Average sim | Sim/SD | Contrib% | Cum% |
|---|---|---|---|---|---|---|---|---|
| *Dosidicus gigas* | Digestive gland | 81.42 | C22:6n3 | 25.01 | 20.87 | 2.04 | 25.63 | 25.63 |
| | | | C16:0 | 20.7 | 17.71 | 7.16 | 21.75 | 47.38 |
| | | | C20:5n3 | 13.42 | 11.11 | 2.01 | 13.65 | 61.03 |
| | | | C18:1n9 | 9.64 | 7.62 | 1.71 | 9.36 | 70.39 |
| | | | C14:0 | 5.7 | 4.79 | 6.18 | 5.88 | 76.26 |
| | | | C18:0 | 4.85 | 4.06 | 5.61 | 4.99 | 81.25 |
| | | | C16:1 | 4.69 | 3.98 | 3.39 | 4.88 | 86.13 |
| | | | C20:1 | 4.49 | 3.66 | 2 | 4.5 | 90.63 |
| | Gonad | 82.46 | C22:6n3 | 25.88 | 22.68 | 4.14 | 27.51 | 27.51 |
| | | | C20:5n3 | 22.04 | 18.31 | 2.47 | 22.21 | 49.72 |
| | | | C16:0 | 20.41 | 17.04 | 5.64 | 20.66 | 70.38 |
| | | | C20:1 | 14.28 | 12.31 | 7.03 | 14.93 | 85.31 |
| | | | C18:0 | 10.93 | 8.59 | 3.6 | 10.41 | 95.72 |
| | Muscle | 83.44 | C22:6n3 | 41.28 | 37.41 | 3.52 | 44.84 | 44.84 |
| | | | C16:0 | 26.4 | 20.9 | 3.17 | 25.05 | 69.89 |
| | | | C20:5n3 | 13.62 | 10.92 | 1.8 | 13.08 | 82.97 |
| | | | C18:0 | 9.32 | 7.72 | 6.59 | 9.25 | 92.22 |
| *Xiphias gladius* | Liver | 78.45 | C18:1n9 | 49.23 | 45.49 | 8.02 | 57.99 | 57.99 |
| | | | C16:0 | 11.89 | 10.26 | 291.78 | 13.08 | 71.06 |
| | | | C18:0 | 9.49 | 8.12 | 242.8 | 10.35 | 81.41 |
| | | | C20:1 | 7 | 5.63 | 2.43 | 7.18 | 88.6 |
| | | | C22:6n3 | 8.32 | 2.22 | 0.58 | 2.83 | 91.43 |
| | Gonad | 66.38 | C18:1n9 | 23.74 | 18.54 | 2.13 | 27.93 | 27.93 |
| | | | C16:0 | 25.16 | 17.85 | 2.41 | 26.9 | 54.82 |
| | | | C22:6n3 | 25.09 | 16.37 | 1.2 | 24.66 | 79.48 |
| | | | C18:0 | 14.58 | 9.16 | 1.79 | 13.8 | 93.28 |
| | Muscle | 89.53 | C18:1n9 | 40.56 | 37.21 | 14.14 | 41.57 | 41.57 |
| | | | C16:0 | 15.54 | 15.3 | 65.34 | 17.09 | 58.66 |
| | | | C22:6n3 | 14.1 | 11.62 | 4.66 | 12.98 | 71.64 |
| | | | C18:0 | 7.41 | 7.01 | 10.68 | 7.83 | 79.47 |
| | | | C20:1 | 5.32 | 4.47 | 5.33 | 5 | 84.47 |
| | | | C16:1 | 4.13 | 3.61 | 4.9 | 4.03 | 88.5 |
| | | | C14:0 | 2.77 | 2.64 | 58.35 | 2.95 | 91.45 |

**Note:**
Av. Sim, Average similarity; Contrib%, Contribution Percentage; Cum%, Cumulative Contribution Percentage.

(C18:3n6) alpha-linolenic acid, ALA (C18:3n3), ARA (C20:4n6); and essentials: EPA (C20:5n3), DHA (C22:6n3)) are acquired through the consumption of whole prey and/or parts of its body rich in these lipid biomolecules. Considering a time period that may vary among species (*Rosas-Luis et al., 2017*; *Zhang et al., 2023*), these lipid components are incorporated conservatively in the tissues of predators, to be subsequently used in various fundamental physiological processes that strongly influence the survival of individuals and

consequently the stability of their populations (*Parzanini et al., 2018*; *Pethybridge et al., 2014*; *Scharnweber, Chaguaceda & Eklöv, 2021*).

At an intra-individual and/or intra-specific level, in our study the comparison of the fatty acid profiles of the analyzed tissues of each species (in jumbo squid: muscle and gonad *vs*. digestive gland; in swordfish: liver *vs*. muscle and gonad) presented some significant differences, while the "intra-individual" comparisons showed no significant differences (PUFA of the gonad *vs*. mantle muscle in *D. gigas*; while in *X. gladius* the muscle and liver in SFA, MUFA and PUFA). These can be explained by the integrative functionality of each organ depending on the role it plays in fundamental physiological processes such as maintenance, reproduction, and growth (*Parrish, 2009*; *Parzanini et al., 2018*). Similar integrative physiology links between organs, functionality and type of stored fatty acid have been described for both highly migratory species *X. gladius* (*Lazo-Andrade et al., 2021*), *D. gigas* (*Gong et al., 2020*; *Quispe-Machaca et al., 2022*), as well as for other bony fish (*Thunnus tonggol*, *Saito et al., 2005*) and chondrichthyans (*Pethybridge, Daley & Nichols, 2011*). In turn, in this interspecific comparison (between the organs of these species), a moderate level of proximity and/or similarity of the fatty acid profile was observed between some jumbo squid and swordfish organs. In particular, the digestive gland of the jumbo squid (prey), an organ rich in precursor and essential PUFAs, presented a similarity with the tissues and/or organs of its predator, mainly with the liver and muscle of the swordfish. According to the ANOSIM results, an adequate similarity was between the digestive gland of the jumbo squid and the muscle and gonad (rather than the liver) of the swordfish. This may indicate that swordfish when consuming large jumbo squids would prefer mainly the digestive gland, which is the nutrient storage organ and contains high amounts of essential fatty acids. This finding may be corroborated by the specialist hunting habit of the swordfish on the jumbo squid (*Ibáñez, González & Cubillos, 2004*), which first stab and cut their prey with the sword of the mouth structure of *X. gladius* (*Berkovitz & Shellis, 2017*; *Preti et al., 2023*), and is then devoured right in this area of the squid body, where large amounts of fats and oils characterized predominantly by PUFAs are found (*Gong et al., 2020*; *Quispe-Machaca et al., 2022*).

Due to (i) the high energetic cost of biosynthesizing PUFA fatty acids in large predatory fish (*Monroig, Tocher & Navarro, 2013*; *Tocher, 2010*; *Twining et al., 2021*), (ii) the availability of these PUFAs in their prey (*Quispe-Machaca, 2019*; *Tan, Zhang & Zheng, 2022*), and (iii) the specialization of their mouth structures to capture only prey with certain energetic attributes (*e.g.*, the swordfish *X. gladius* slashes the jumbo squid *D. gigas* with its sword and mainly consumes its digestive gland as a big size prey rich in fats) (*Loor-Andrade et al., 2017*; *Preti et al., 2023*; *Lazo-Andrade, Barría & Urzúa, 2024*), we propose a trophic context of acquisition of essential fatty acids from the jumbo squid (prey) and then their storage and mobilization dynamics in a conservative way in the organs of the swordfish (predator). Our results, according to the analysis of similarity of the fatty acid profiles of the digestive gland of the jumbo squid and the organs of swordfish, could reveal as a testable hypothesis in future studies, the pathway use and storage sequence of these FAs (*i.e.*, first processed in the liver, then stored in muscle, and finally transferred to the gonad of swordfish).

Identifying this storage sequence of FAs may be key to the functioning of swordfish since they are used in different physiological functions. Here, the predominant fatty acids in the muscle (SFAs, MUFAs) can be used as an energy source during periods of food absence or scarcity (*Wang et al., 2019*), while essential long chain PUFAs (EPA (C20:5n3), DHA (C22:6n3)) are key to the overall success of the reproductive process from the maturation of the gonad to the formation of the embryo and larva (*Twining et al., 2021*). A similar response has been described for species of teleost fish from the Humboldt Current system (*Rincón-Cervera et al., 2019*). In turn, according to our results on the dynamics of the FAs in jumbo squid organs, we propose that these are first stored in the digestive gland and then transferred to other organs. This dynamic of fatty acids for jumbo squid has also been reported in the winter (*Guzmán-Rivas et al., 2021*) and spring (*Gong et al., 2018*, *2020*) off the coast of Chile so coast of Peru has been reported in winter (*Gong et al., 2020*) and spring (*Gong et al., 2018*; *Saito, Sakai & Wakabayashi, 2014*). On the other hand, FAs (16 carbon PUFA, 18:4n3, 22:4n3, 22:5n3), as found in previous studies on highly migratory fish (*Guzmán-Rivas et al., 2023*; *Lazo-Andrade et al., 2021*), were detected in traces (*i.e.*, in very small amounts). A plausible explanation for the low presence of these FAs in the analyzed samples may be due to the role (highly dynamic) that these FAs play in the biosynthesis of long-chain highly unsaturated fatty acids, described through the complex metabolic pathway of Sprecher (for details of the model see: *Monroig et al., 2022*; *Tocher, 2003*).

In the context of energy reserve dynamics, seasonal differences in FAs found in the organs of the studied species have also been reported in previous studies (*Quispe-Machaca et al., 2021*, *2022*; *Lazo-Andrade et al., 2021*). The findings revealed by the present study showed the same trend for this time of year (winter), but with some slight variations between consecutive years (2021, 2022). Given that the study area (South Eastern Pacific Ocean: SEPO) presents pronounced seasonal variations in oceanographic conditions (temperature, upwelling) that may modulate the availability of food and/or prey in the environment (*Ibáñez et al., 2021*; *Lazo-Andrade et al., 2023*), and consequently, influence the capacity of these species to accumulate lipid reserves, future comparative studies should expand the time window of the analyses to include more seasons of the year (spring-summer). Also, due to the logistical limitations of the study, including the sampling event and restrictions on capturing specimens, future research should complement our findings and observations (*i.e.*, swordfish eating jumbo squid), considering techniques (isotopes, barcoding, *etc.*) that can reveal long-standing feeding habits (seasonal, over years) (*Günther et al., 2021*; *Lovell et al., 2024*; *Olinger et al., 2024*).

Provided that predator-prey interactions between species occur at the spatial/temporal level of both of the evaluated species in the SEPO, the trophic migration described for swordfish (from 50°N to 50°S, *Preti et al., 2023*) may coincide spatially and temporally with the reproductive migration of the jumbo squid in this area of the SEPO (*Castillo et al., 2020*; *Ibáñez, González & Cubillos, 2004*; *Ibáñez et al., 2015*). Considering this background, a trophic cascade effect could be generated, in which the prey consumed by jumbo squid is finally reflected in the fatty acid profile of swordfish. Within the spatiotemporal context of trophodynamics in the marine food web, future studies should prioritize understanding

current trophic dynamics, particularly the transfer and storage of fatty acids in predator-prey interactions. Subsequent research should then examine how the feeding habits (*i.e.*, prey selection) of species in the Humboldt System have shifted under different climate and oceanographic conditions, such as El Niño Southern Oscillation (ENSO) and its phases (*Bertrand et al., 2020*; *Lazo-Andrade, Barría & Urzúa, 2024*). Additionally, it is essential to assess how the phenology of these species may have been altered, as discussed in match-mismatch theory (*Durant et al., 2013*; *Ferreira, Neuheimer & Durant, 2023*), within the context of climate change. In this context, the phenomena of climate change in marine environments is characterized by an increase in water temperature and hypoxia events, which can modulate variations in the distribution range of species, as well as their density; and consequently, generate changes in the predator *vs.* prey interactions that occur in the Humboldt Current System (*Assan et al., 2020*; *Draper & Weissburg, 2019*; *Laws, 2017*).

## CONCLUSION

Finally, our findings on the use of fatty acids as biomarkers of the interaction between two highly migratory resources of the Humboldt System may reveal a moderate degree of preference of swordfish in preying on jumbo squid, where precursor fatty acids predominate (*i.e.*, ALA (C18:3n3) and ARA (C20:4n-6)) along with essential long chain PUFAs (EPA (C20:5n3), DHA (C22:6n3)) for their growth and reproduction. We conclude that this feeding strategy (preference on jumbo squid) and integrative physiological strategy of incorporating specific fats for their subsequent use in different organs for fundamental physiological processes (growth and reproduction) could elucidate a possible convergent physio-energetic strategy in the lipid storage and use of essential biomolecules present in species of higher vertebrates considered top predators in their habitat (marine or terrestrial, as appropriate). Thus, further research is needed to fully understand the extent of these convergent strategies. As mentioned above, this strategy has been widely described for top predators in the food webs of terrestrial environments, but scarcely for top predators in the marine food web; therefore, our study is a pioneer in revealing this type of trophic interaction between two top marine species in the Humboldt Current System.

## ACKNOWLEDGEMENTS

We really thank the editor and reviewers for their constructive criticism and important suggestions. Special thanks to Christine Harrower for correcting the English and improving this manuscript. We thank the scientific observers of the Instituto de Fomento Pesquero (IFOP) for assistance aboard fishing vessels.

### Funding

This work was funded by the Instituto de Fomento Pesquero (IFOP). The "Fisheries Project of Highly Migratory Resources" supported this work: grant 581136 (Biological-

Fishing Aspects, IFOP). The funders had no role in study design, data collection and analysis, decision to publish, or preparation of the manuscript.

## Grant Disclosures

The following grant information was disclosed by the authors:

Instituto de Fomento Pesquero (IFOP).

The "Fisheries Project of Highly Migratory Resources" (Biological-Fishing Aspects, IFOP): 581136.

## Competing Interests

The authors declare that they have no competing interests.

## Author Contributions

- Marco Quispe-Machaca conceived and designed the experiments, performed the experiments, analyzed the data, prepared figures and/or tables, authored or reviewed drafts of the article, and approved the final draft.
- Fabián Guzmán-Rivas conceived and designed the experiments, performed the experiments, analyzed the data, prepared figures and/or tables, authored or reviewed drafts of the article, and approved the final draft.
- Patricio Barría Martínez conceived and designed the experiments, authored or reviewed drafts of the article, and approved the final draft.
- Christian Ibáñez conceived and designed the experiments, authored or reviewed drafts of the article, and approved the final draft.
- Ángel Urzúa conceived and designed the experiments, performed the experiments, analyzed the data, prepared figures and/or tables, authored or reviewed drafts of the article, and approved the final draft.

## Data Availability

The raw measurements are available in the Supplemental Files.

## Supplemental Information

Supplemental information for this article can be found online at http://dx.doi.org/10.7717/peerj.19129#supplemental-information.

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
