# Peer review of "Fatty acid biomarkers reveal the interaction between two highly migratory species in the Southern Humboldt System: the swordfish and its prey, the jumbo squid"

_PeerJ, doi:10.7717/peerj.19129_

## Round 0.1 · original submission · Major Revisions

Please carefully review the comments from three experts. There are major concerns pertaining to the statistical analysis and sampling methodology that need to be thoroughly addressed in a revised version and rebuttal letter.

**Language Note:** PeerJ staff have identified that the English language needs to be improved. When you prepare your next revision, please either (i) have a colleague who is proficient in English and familiar with the subject matter review your manuscript, or (ii) contact a professional editing service to review your manuscript. PeerJ can provide language editing services - you can contact us at [email protected] for pricing (be sure to provide your manuscript number and title). – PeerJ Staff

Reviewer 1 ·

Basic reporting

The article is well-written and interesting to read, its structure is coherent, and it represents an appropriate unit of publication. Suggestions to improve the manuscript:

The introduction is concise, it gives a clear context about the topic. However, the author should add more information on fatty acid trophic markers, how they have been used for studying marine aquatic environments, and how their study is useful in understanding the trophic ecology of the swordfish and the jumbo squid.

More literature could be referenced on fatty acid trophic markers and fatty acids and trophic interactions to gain a better understanding of the meaning of this study. For example: https://doi.org/10.3389/fmars.2023.1132246, https://doi.org/10.7717/peerj.12336, https://doi.org/10.1007/s00300-020-02788-y

Some of the information on the profiles of fatty acid composition could be presented as images (bar charts).

Experimental design

The manuscript is within the scope of PeerJ, it is an original research, and its aim was clearly defined.

The authors must explain why was a non-parametric test chosen (ANOSIM) rather than a parametric test (PERMANOVA)? How were normality and homogeneity of variances analyzed?

Validity of the findings

The authors should clarify the issue related to the choice of statistical tests used.

The results presented in lines 293-294 are different from those presented in Table 3.

The discussion section should also include the limitations of the study. In this way, since fatty acid biomarkers provide indirect evidence of trophic interactions (it is not possible to discriminate which prey is being ingested by the swordfish), the authors may also suggest other types of analysis that could support their assumptions (for example, fatty acids profiles of other prays, isotopes, barcoding, etc.)
.

Additional comments

Other suggestions/comments:

Line 125: X. gladius instead of Xiphias gladius (the genus was already written in full in line 86)
Line 147: HCS instead of Humboldt Current System.
Line 177: Specify the type of muscle that was sampled (mantle muscle?)
Line 212: “(PCoA). The fatty” instead of “(PCoA) The fatty”
Line 213: “PCoA” instead of “principal coordinate analysis”
Line 215: What was the significance level established? (0.05?)
Line 266: “The n3 PUFA” instead of “the n3 PUFA”
Line 295: (Table 3) instead of “(Table 4)”
Line 341: some of the “intra-individual” comparisons do not have significant differences (Gonad-Xg vs Muscle-Xg, Liver-Xg vs Muscle-Xg, Gonad-Xg vs Liver-Xg).
Line 346: X. gladius instead of Xiphias gladius and D. gigas instead of “Dosidicus gigas” (the genera were already written in full)
Line 352: According to ANOSIM results, the highest similarity was with muscle and gonad (rather than of liver).

·

Basic reporting

The manuscript is well written, with relevant results that support its hypothesis—no more comments to add.

Experimental design

No comments.

Validity of the findings

No comments.

Additional comments

My main concern with this manuscript is in the Discussion section. The authors do not mention the temporal FA changes in tissues of Dg and Xg that they have seen in past articles published by them, like Quispe-Machaca et al. (2022; https://doi.org/10.1016/j.fishres.2021.106154) and Guzmán-Rivas et al. (2023; DOI 10.7717/peerj.15524). Also, consider what the authors state in lines #387-394, where they mention that understanding the trophodynamics in the marine food web is the basis for understanding climate-oceanographic scenarios, with no further explanation. The authors should extend their discussion and discuss what they have found in past articles to construct a more explicit statement. They need to explain the Dosidicus gigas FAA results when considering its prey, as was reported by Quispe-Machaca et al. (2022; https://doi.org/10.1016/j.fishres.2021.106154).

Reviewer 3 ·

Basic reporting

See below
There was no hypothesis stated.

Experimental design

It was opportunistic sampling.

Validity of the findings

See below

Additional comments

This paper is set up as an investigation of trophic relationship between swordfish and squid, using FA in various organs of the predator and prey. The introduction talks about the importance of understanding trophic ecology in general and specifically the relationship between these two organisms. However, the authors reveal halfway through the Introduction they know quite well through observation that the swordfish specifically targets the digestive gland of the squid. If they know that, there really isn’t any additional information to be obtained by comparison of FA in predator and prey, or their organs. The authors argue that FA composition can provide information about ‘their dynamics’ and the ‘potential physiological role that lipids and fatty acids play’, presumably in the predator. They also suggest that the comparison could provide info on the rates of assimilation and transfer and storage. Unfortunately, with only compositional data from opportunistically collected samples, they can’t provide any of this information. There are also issues with their statistical approach, presentation of results and their Discussion. I suspect that this data could have made a solid contribution to the author’s earlier paper reporting the targeting of squid digestive gland by swordfish. However, there is not enough information here for this to be a stand-alone contribution. Perhaps if the authors had spatial or temporal variation, there might be interesting patterns to report.

Specific comments:
The introduction and discussion consist of several numbered lists with very little context. The authors need to explain why the elements in these lists are important and relevant for the work they are proposing.

Lines 90-97: This is a 2-sentence paragraph. One of the sentences is just a list of types of studies involved in trophic ecology.

Lines 145-146: States that the objective of the study was to use FA to evaluate the interaction of the two species but they already know how they interact from observation. Perhaps a more interesting aim would be to evaluate the use of FA as appropriate tracers in this system?

Line 160: What do the authors mean by ‘put to sleep’? Euthanized would likely be more accurate.

Line 166: If n=31, why are there only 3 data points for liver and muscle of X. gladius in Fig. 1?

Line 180: Were samples homogenized before drying?

Lines 186-188: 20 mg dry weight samples were extracted. Why so little? What was the ratio of dichloromethane to methanol? If only 4 ml of KCl was use (KCl is correct, not KCL), this is not a Folch et al extraction and lipids would not have been quantitatively extracted. Far more solvent would be needed to fully extract non-polar lipids.

Lines 199-200: FA were measured with GC at a set temperature? Does that mean isothermal? If so, not all FA can be resolved in a marine sample, leading me to question the accuracy of the data. As it stands, a number of common marine FA have been omitted from the data set, including 16 carbon PUFA, 18:4n-3, 22:4n-3, 22:5n-3.

Line 204: What internal standard was used?

Lines 214-217: I see that ANOSIM is likely the better technique to use here since PerMANOVA would simply reveal that all FA proportions were different, while we can more easily interpret the results of ANOSIM using the R values. However, I do not see what contribution SIMPER makes. Why would we be interested in the similarity within a sample type?

Results: In general, these are just, literally, restating the data in their tables. It would be far better to summarize and describe patterns. Or simply draw our attention to the more important aspects of the FA data.

The first paragraph of the Results just tells us the content of each table. That is not necessary. Just tell us about the data.

The comparison of the FA profiles of the two species (the most important section) also just quotes the ANOSIM results table. A summary of important points (perhaps that lowest R values are found for FA comparisons of pred tissues with the prey’s digestive gland, indicating greatest similarity) would be more useful.

Discussion: It’s repetitive and doesn’t really add anything. It’s more of a review of use of lipids in trophic ecology. Unfortunately, with the very thin dataset, there isn’t a lot the authors can say.

---

## Round 0.2 · Major Revisions

Significant challenges remain in the manuscript. Please resolve these at your earliest opportunity and submit a markedly enhanced version at your convenience.

Reviewer 1 ·

Basic reporting

The authors worked on the suggestions made by the reviewers, but there are still areas for improvement before the manuscript can be published.

* Even though it is mentioned in the response letter that a native speaker reviewed the document, there are still grammatical aspects to be corrected. For example:
L2: Fatty acid biomarkers reveals
L47-48: However, regarding how the fatty acid profile varies in the various organs of the predator and its prey are still unknown
L54: by the consume of digestive gland
L83: iii) omnivores that consume both seaweeds, invertebrates and fishes

* Some sentences need to be rewritten for clarity. For example:
L30: The current phrasing states that organisms tolerate exposure to multiple environmental stressors, such as temperature, oxygen, and salinity.' To enhance clarity, I would suggest specifying that these factors are only considered stressors when they fall outside the organism's physiological tolerance limits.
L142-152: The authors mention the information about the swordfish in parentheses in the first sentence, which could make the reference to the swordfish in the second sentence sound somewhat disconnected. To improve the text's cohesion, avoid using parentheses for the first mention of the swordfish.

* The introduction seems quite long (approximately 1570 words). It includes basic concepts of marine trophic ecology and examples (L76-89) that are informative and valuable but may not be necessary for this study. A more concise introduction would be preferable, focusing on the critical ecological background directly related to the authors' analysis, such as the jumbo squid's and swordfish's trophic roles and their relevance to fatty acid transfer.

* The authors provided a summary paragraph of all the results at the beginning of the Results section. Since the subsequent subsections already cover the detailed findings for the squid profile, swordfish profile, and their comparison, it might not be necessary to include this summary. Alternatively, if you prefer to keep it, I suggest limiting it to key points that are not repeated in the following subsections.

Experimental design

The procedure for obtaining lipids could be more precise since, in L228-242, the method is described twice but with minor modifications. I suggest putting this information into a single description.

The term 'cuttlefish' was used in L298-299, but since the fatty acid analysis was conducted on squid and swordfish, the use of 'cuttlefish' seems to be a mistake.

Validity of the findings

* L409-419: Regarding the interpretation of the lipid profiles, it's important to note that the incorporation of dietary fatty acids into the predator's tissues is a process that takes time. Therefore, if samples were taken shortly after predation, the observed fatty acid profiles might not fully represent the predator's long-term dietary intake.

* L432-434: While the author mentions a trend of similarity in the fatty acid profiles between organs of D. gigas and X. gladius, the R ANOSIM values suggest a more complex picture. Specifically, the value for the digestive gland (R=0.57) indicates a moderate level of similarity, and the R values for the gonad (0.99) and mantle (0.97) show very high levels of dissimilarity, suggesting differences rather than similarities. Therefore, the conclusion of a general trend of similarity between these species' organ profiles might be overstated.

*L450-455: While the analysis of fatty acid profiles provides insight into these fatty acids' potential storage and mobilization, the proposed pathway (processed in the liver, stored in muscle, and transferred to the gonad) should be considered a hypothesis.

Additional comments

• The authors mention D. gigas as a 'top predator' in the manuscript. Could you clarify the relevance of this description? Since, in this study, the jumbo squid is considered the prey of X. gladius and not a predator, I find the reference to its position as a 'top predator' confusing. If this characterization is important to the study, could you please explain its role more clearly?
• I suggest mentioning the predator and prey identification only when necessary, as this is already clearly established in the introduction.
• L208-211: In the Methods section, there is a sentence that provides information about previous studies. Therefore, it is more appropriate to place it in the Results or Discussion section.
• If the methodology used follows the modification by Cequier-Sánchez et al. (2008), is it worth mentioning the updates by Guzmán-Rivas et al. (2023), Quispe-Machaca et al. (2021, 2022), Lazo-Andrade et al. (2021, 2023)?
• L382: The heading "SIMPER" might not be appropriate as it refers to the statistical technique used rather than the results or biological insights derived from the analysis.
• L449-450: The claim of specialization in the swordfish's mouth structure to capture only prey with high lipid content requires further evidence.
• L495-498: While studying the potential impacts of climate change on feeding behaviors and fatty acid profiles of species in the Humboldt system is undoubtedly essential, in terms of this investigation, it might be more critical to first focus on understanding the current trophic dynamics and how fatty acids are transferred and stored within predator-prey interactions.
• L515: it would be helpful to clarify that while some similarities in lipid storage and use might exist, further research is needed to fully understand the extent of these convergent strategies.

---

## Round 0.3 · Minor Revisions

Please make the minor corrections suggested by the reviewer.

Reviewer 1 ·

Basic reporting

The authors have addressed the corrections and recommendations provided by the reviewers. In this regard, I suggest that the manuscript can proceed to publication. However, I recommend that it undergo one final review to address English language issues. For example:

A) “Consequently, some degree of similarity in the FA profiles found in the tissues of the predator and its prey are expected.” - the correct verb form is "is expected" rather than "are expected".
B) “As part of monitoring program of “Fisheries Project of Highly Migratory Resources…” - “As part of the monitoring program of “Fisheries Project of Highly Migratory Resources…”.
C) “… after which 4 mL of potassium chloride (0.88% KCL in ultra-pure water) were added and centrifuged at 1500 RPM for 5 min” - the correct verb form is "was added" rather than "were added".
D) “In brief, 1 mL of the lipid extract were esterified by incubating” - the correct verb form is "was esterified" rather than "were esterified".

These grammar issues are minor enough that I would not need to re-review the manuscript once the corrections are made.

Experimental design

no comment

Validity of the findings

no comment

Additional comments

no comment

---

## Round 0.4 · accepted · Accept

Thank you for answering the questions. Your manuscript has been accepted.